# Representation of Whom? Ancient Moments of Seeking Refuge and Protection

Elena Isayev

Department of Classics and Ancient History, University of Exeter, Amory Building, Rennes Drive, Exeter EX4 4RJ, UK; e.isayev@exeter.ac.uk

**Abstract:** Within the ancient corpus we find depictions of people seeking refuge and protection: in works of fiction, drama and poetry; on wall paintings and vases, they cluster at protective altars and cling to statues of gods who seemingly look on. Yet the ancient evidence does not lend itself easily to exploring attitudes to refugees or asylum seekers. Hence, the question that begins this investigation is, representation of whom? Through a focus on the Greco-Roman material of the Mediterranean region, drawing on select representations, such as the tragedies *Medea* and *Suppliant Women*, the historical failed plea of the Plataeans and pictorial imagery of supplication, the goal of the exploration below is not to shape into existence an ancient refugee or asylum seeker experience. Rather, it is to highlight the multiplicity of experiences within narratives of victimhood and the confines of such labels as *refugee* and *asylum seeker*. The absence of ancient representations of a generic figure or group of the 'displaced', broadly defined, precludes any exceptionalising or homogenising of people in such contexts. Remaining depictions are of named, recognisable protagonists, whose stories are known. There is no 'mass' of refuge seekers, to whom a single set of rules could apply across time and space. Given these diverse stories of negotiation for refuge, another aim is to illustrate the ways such experience does not come to define the entirety of who a person is or encompass the complete life and its many layers. This paper addresses the challenges of representation that are exposed by, among others, thinkers such as Hannah Arendt, Liisa Malkki and Gerawork Gizaw.

**Keywords:** displacement; forced migration; representation; refuge; asylum; Aeschylus; supplication; Greek Tragedy; Cassandra; Aeneas; exceptionalism; Euripides; *Herakleidai*; *Medea*

## 1. Introduction

The question that begins this investigation is simply, representation of whom? Such a starting point is necessary because the ancient evidence does not easily lend itself to exploring attitudes to refugees or asylum seekers, either individually or in groups. At least this appears to be the case in the Greco-Roman material of the Mediterranean region in the period which will be the focus here—before the Roman Empire of the 1st century CE. The situation changes somewhat towards Late Antiquity and with the coming of Christianity.[1] The reason that representation and attitudes are difficult to gauge is not that there were few who sought refuge and protection. Numerous records attest to people being driven from their homes because of warfare, the threat of violence, enslavement, political instability, social tensions and environmental pressures. However, people who had to endure such challenging circumstances were not envisioned as necessarily belonging to a distinct category in these records. The multiple factors which defined their perilous condition were not flattened into a single entity that could form the target of policies and particular attitudes, or be the subject of study and representation—whether that is in literary works, historical narratives, paintings or sculpture. Nor did such a condition define the entirety of the person or designate their status.

The predicament that in the 21st century is often referred to as 'forced migration' in ancient contexts is not separated from other challenging experiences that result in people

having to endure precarity, exclusion, violence and the lack of protection or a home for other reasons. What is more, the experience of exclusion or exile is often not the focus of the accounts within which it is conveyed but rather is intertwined with diverse narrative strands that highlight different sociopolitical concerns. Authors have used exchanges between refuge seekers and potential hosts as a poignant setting and catalyst for discourses on ethical dilemmas concerning democracy, cosmopolitanism, sovereignty, hospitality and other subjects relating to civic society and foreign affairs.[2]

In reflecting on select ancient representations in text and image, the goal of the exploration below is not to shape into existence an ancient refugee or asylum seeker experience. Rather, it is to highlight what may be lost by pressing a multiplicity of experiences within narratives of victimhood and the confines of such labels as *refugee*, *asylum seeker*, and even perhaps the *displaced*. The persistent challenges of moving beyond such representations in the 21st century are captured in Gerawork Gizaw and Kate Reed's reflections on Alideeq Osman's autobiographic *Prison of Dust*: 'woven through his memoir is an ambivalence about this project of representation as an impetus for empathy, and empathy as an impetus for action. "There are no words," he admits, "to describe the real fear and horror inside each thirsty and hungry person that showed up, seeking refuge during the Somali civil war"' (Gizaw and Reed 2022; Osman 2020). The goal in what follows is thus to illustrate the ways that such experience, even if it persists across a whole lifetime and transforms into permanent temporariness, still does not define the whole of who a person is or encompass the complete life and all its layers.[3] Ancient voices help give meaning to Hannah Arendt's *We Refugees*: 'In the first place, we don't like to be called "refugees". We ourselves call each other "newcomers" or "immigrants"'.[4] As Arendt traces the multiple routes, externally and internally taken by those who were persecuted by the Nazi regime, she outlines the instability of identity and how there is no single 'We'.

Those representations of ancient suppliants or refuge seekers which remain are never generic but are rather of named people whose stories are known and whose lives encompass moments of such displacement and precarity. As we will see, they are often the protagonists of myths and are depicted in scenes from epics and well-known tragedies: such powerful characters as Medea and the Trojan war hero Aeneas. We will examine the way that negotiations for refuge with potential hosts become the setting for Classical Greek tragedies of the 5th century BCE, as Aeschylus's *Suppliant Women* and Euripides' *Herakleidai*. Unlike the mythical corpus, there are few historical records that chronicle in any depth requests for protection. Of these, the most extensive from the period which we will consider is that of the Plataeans, who came to Athens for refuge following their city's takeover by the Thebans in the 370s BCE.[5] Most other historical references to refuge seekers appear only in passing, usually within ancient accounts of war and conquest. These often mention sieges and the necessity for local populations to flee in order to escape violence, starvation and enslavement,[6] such as during the Punic Wars between Rome and Carthage in the 3rd century BCE. Polybius, in the first two books of his *Histories,* chronicles the extent of these operations through the numbers of those affected: the 50,000 suffering from famine in the besieged Agrigentum held by Carthaginians in 262 BCE;[7] the 20,000 enslaved by Rome on the capture of Aspis in Libya;[8] and the 10,000 prisoners taken by Rome to Lilybaeum.[9] Such summary notations give few details or commentary on the circumstances under which refuge was sought, given or denied. Historical narratives and epigraphic texts, inscribed on stone or metal, may also include remarks on individuals who were being exiled and the situation that would have led to their expulsion or opportunity for return. We have a number of surviving decrees and honorary stelae with the names of exiles inscribed along with those of their exilic hosts.[10]

The most-in-depth explorations from within moments of exile are given by those experiencing it, but even these figures have little to say about seeking refuge or the negotiations to obtain it. For the exiled philosopher, Diogenese the Cynic, from Sinope, who lived in Athens for part of his exile during the 4th century BCE, the focus of his philosophical expositions was to subvert his destitute condition, using it to critique the inequities of

the polis-state world that he inhabited (Gray 2015; Isayev 2021). In the writings of other prominent figures who experienced exile, such as the 4th-century BCE Greek general and historian Xenophon or the later Roman politician Cicero and the Latin poet Ovid, the focus of their emotion is not the search for refuge but the longing for home, which remains inaccessible.[11] Hence, for the purpose of this exploration, such exiled figures will be set aside because I have had a chance, elsewhere, to consider some of them within the context of ancient wandering and permanent temporariness (Isayev 2017b, 2021, 2022). Here, the focus will be on the few surviving ancient representations of people in the moment of seeking refuge and protection, as distinct from those who wrote from exile, having found somewhere to inhabit but who were unable to return to a place of home.

As the following discussion will show, the specificity of each context within which ancient negotiations for refuge took place seems to preclude the emergence of a generic suppliant case or the manifestation of a homogenised 'mass' of refuge seekers to whom a single set of rules could apply across time and space.[12] What is not clear from the ancient evidence is what proportion of those seeking refuge would have needed official permission from authorities to physically remain at a particular site, as distinct from any request to gain some form of civic status or recognition that would bring with it rights and responsibilities. In a world without any equivalent to the territorial nation-state, where the mere physical presence within its borders activates certain rights and the potential for status and belonging, thus prompting an increasingly securitised border industry, those in need of refuge in the world of ancient city-states could physically access most places, except for the site or region where they were expelled from or where their well-being was threatened.[13]

There are the rare occasions when the origin-polis (city-state) forbade asylum to be given to its own citizens. This was the case of the Acarnanians, who in desperation during the Aetolian invasion in 211 BCE, passed a severe resolution: if any Acarnanians survived and escaped, authorities beseeched that no one was to give them refuge, and they cursed all who took fugitives into their territory.[14] Technically, the challenge for the ancient refuge seeker was primarily one of the inability to return to a designated home place rather than the inability to find a new one. This does not mean that there would be a welcome at every port,[15] not least because those seeking refuge one day may have been the perpetrators of former violence and expulsion. In addition, there would have been particular pressures when whole communities found themselves homeless owing to siege and conquest for extended periods of time. The intricate negotiations to gain protection in the cases considered here were necessary because the requests were not just for the ability for those who fled to physically remain where they were, but also for the commitment by the host to take up arms to repel any who may threaten the refuge seekers and potentially the host-community itself, for its decision to grant refuge.

## 2. Protagonists of Ancient Refuge Seeking

Primarily, ancient depictions of refuge seeking and supplication are in the form of testimony given directly by those who are in search of it. This is the case for the mythical characters in legendary and dramatic accounts who speak on their own behalf. Of those with a particularly strong voice is Medea as she is portrayed in Euripides's tragedy of the same name created in 431 BCE.[16] This mythical figure, who helped Jason and the Argonauts acquire the Golden Fleece at the cost of her home and family, appears in multiple guises through the ages. She is a complex character who gains sympathy despite her horrific crimes, which are driven by the intrigues of gods and men; although she is the victim of their machinations, she refuses to be confined by victimhood.

> But will you banish me without the regard due a suppliant?
>
> . . .
>
> I accept my exile: it was not exile I sought reprieve of.
>
> (Euripides, *Medea* lines 326, 338)

> What do I gain by living? I have no fatherland, no house, and no means to turn
> aside misfortune. My mistake was when I left my father's house, persuaded by
> the words of a Greek. This man—a god being my helper—will pay for what he
> has done to me.
>
> ... Let no one think me weak, contemptible, untroublesome.
>
> No, quite the opposite, hurtful to foes, to friends kindly. Such persons live a life
> of greatest glory.
>
> (Euripides, *Medea* lines 798–810)

Euripides's harrowing dramatisation of Medea's abandonment by the oath-breaking Jason, of her wrath, of the agony of loss and of her enduring exile, continues to resonate through the ages.

Other prominent voices from moments of refuge seeking are those of the mythical Danaids, as portrayed in Aeschylus's *Suppliant Women*. This tragedy, first performed in the 460s BCE, is an exploration of the balance of power and responsibility at the start of the Athenian experiment with polis-state democracy (see note 16). Its protagonists are the chorus of 50 daughters of Danaeus who left Egypt for Argos to seek refuge from forced marriages with their cousins, who were in pursuit. The Danaids' position appears very different from Medea's, and their request for refuge is more straightforward. However, there is affinity in the root causes of their exile: the consequence of their resistance to being made pawns in a world driven by the ambitions of gods and men. They too, like Medea, refuse to be treated as mere victims of their fate. Even when represented as a group, as here, those seeking refuge are envisaged as individuals, who may have a similar story but who do not form a 'mass'. Thus, although they speak as a chorus, they also use 'I', the Greek ἐγώ, in reference to themselves:[17]

> So I (ἐγώ) too, fond of lamenting in Ionian strains,
>
> rend my soft, sun-baked cheek
>
> and my heart unused to tears;
>
> I cull the flowers of grief,
>
> in apprehension whether these friendless exiles
>
> from the Land of Mists
>
> have any protector here.
>
> (Aeschylus *Suppliant Women*, lines 69–76)

Before we explore further these Suppliant Women's positioning of themselves in negotiations for refuge, it is worth briefly pausing on the well-known Greco-Roman epics, which are often drawn on for considering the condition of exile and wandering through the stories of central figures.[18] These include Homer's *Odyssey*, the 8th-century BCE account of the long journey home from the Trojan battle of one of its Greek heroes, Odysseus, and Vergil's *Aeneid*, written some 700 years later, in the 1st century BCE, chronicling the journey stories of Aeneas, another of the Homeric heroes, though from the side of the besieged city of Troy. Vergil's epic extends the story by charting Aeneas's flight from his burning city and eventual arrival in Italy with his followers, thus stitching his wanderings into the foundation story of Rome. The travails of both of these heroes, Odysseus and Aeneas, are used to represent the experience of longing for home and family, either lost or far away, and the precarity of the wandering condition.There are no epics centred on a female hero from this period, a subject beyond the scope of this paper, but it is important to acknowledge the prominence of stories with female protagonists such as Medea or Helen (both the focus of tragedies by Euripides) who in very different ways embody many of the sentiments of exile and exclusion.[19]

Another epic incorporating wandering is *Exodus*, the second book of the Bible, which is important to note here for its different approach and focus, not on the individual life story but rather on that of a whole people. It too is based on traditions of the 8th century

BCE, even if those go beyond the immediate shores of the Mediterranean. While there is some emphasis on prominent figures, in this case divinely chosen leaders such as Moses and those around him, it is among many things primarily the story of a 'nation' whose journey is not only towards place but freedom and the divine. In the Greco-Roman epics and myths, there is no similar wandering narrative of a whole people. Even if there are instances of groups who claim kinship with their potential hosts and in relation to certain regions—as do the Suppliant Women, who fleeing from Egypt assert their common descent with the Argives through the line of the mythical Io. These legendary stories are distinct from the way they are used in later claims to lands, cities and resources through story making of past expulsions, and case making for rightful returns, such as those charted in the case of the Messenians (Luraghi 2008).

Ancient historical writings contain scattered references to crises that drive populations to flee their cities and arrive at varied destinations to await a moment of return or to stay away in perpetuity. A well-known example is of the Romans' taking shelter in Veii and other nearby towns when the Gauls sacked their city in 390s BCE. The most extensive surviving account is by the ancient historian Livy, who narrates the events of the Gallic Sack some three centuries later, in the fifth book of his annals of Rome, entitled *Ab Urbe Condita*.[20] Although his description is detailed enough to include fictionalised speeches and debates, there is little about the moment of the Romans' arrival among neighbouring regions and there is no discourse on how refuge was attained. In this episode, the absence of the inhabitants from Rome becomes notable primarily on their return to the city following the departure of the Gallic invaders. Otherwise, little is mentioned of the Romans' experience while in exile. The subject of those who left Rome during the sack surfaces in the narrative, when measures have to be brought in to entice those who fled to seek refuge to return and make decisions about what to do with the destroyed city.

It is notable that any generalisations made by Livy in reference to Rome's population tend to be in relation to the citizens of plebeian social status. They are frequently presented as a homogenised group in Roman historical narratives and often negatively, with particular political interests and struggles for power.[21] Their time away from Rome during the siege, along with that of other citizens, is of little interest to the narrator, who also does not refer to them as either refugees or exiles, upon their flight from the city. This may be compared with the multiple references to the pain of being expelled from Rome experienced by the Roman general Camillus during his exile in Ardea, at the behest of Roman authorities prior to the Gallic Sack: 'Camillus was languishing there in exile, more grieved by the nation's calamity than by his own' (Livy 5.43.7–8). He was eventually recalled from exile (Livy 5.46) to help defeat the Gauls and became the heroic figure who reclaimed Rome for the citizens.

Such vivid portrayals of return, as of the fictionalised Roman return to their city described here, are rare. Primarily, what remains in historical narratives are passing references within virtually formulaic depictions of events in the wake of conflict. Such a trope is evident in Livy's description of the imagined 7th-century BCE Alban departure from their destroyed city, which focuses on the grief of leaving rather than arrival in Rome and their subsequent settlement there, depicted as usefully helping to increase early Rome's population:

> . . . at Alba oppressive silence and grief that found no words quite overwhelmed the spirits of all the people; too dismayed to think what they should take with them and what leave behind, they would ask each other's advice again and again, now standing on their thresholds, and now roaming aimlessly through the houses they were to look upon for that last time.

> Rome, meanwhile, was increased by Alba's downfall. The number of citizens was doubled.

> (Livy 1. 29.3; 1.30.1)

Such brief accounts are situated as stages within wider historical descriptions of conflict and interstate relations where there is little space for, or interest in, pausing to consider what

such an experience entailed in negotiation for protection, the places of refuge, everyday life there and modes of survival.

## 3. Episodic Images of Flight and Appeals for Protection

Here, we return to the question from the introduction: representation of whom? The aim of outlining some of the mythical and historical figures is to highlight the centrality of specific individual lives in the ancient tradition. Experience of precarity triggered by exile and loss of home, some resulting in extensive wandering or displacement, are encapsulated in narratives of its endurance and struggles for its overcoming. Popular episodic images that appear across historical periods of some of these figures attest to the power of their stories, which capture a shared understanding of the pain of such circumstances. The focus in the exploration that follows is on the choice of scene for representation rather than on its aesthetic quality, which would require further study beyond the scope of this investigation. Of the well-renown episodes depicted, one of the most prolific is a stock image of Aeneas carrying his elderly father, Anchises, on his shoulders in their escape from the defeated burning city of Troy. One example can be found on an Archaic Attic amphora of the late 6th century BCE (Figure 1) This depiction predates by a half a millennium Vergil's retelling of the episode in the 1st century BCE in his second book of the *Aeneid*: [22]

Come then, dear father, mount upon my neck;

on my own shoulders I will support you, and this task will not weigh me down.

(Vergil *Aeneid*, 2.707–708)

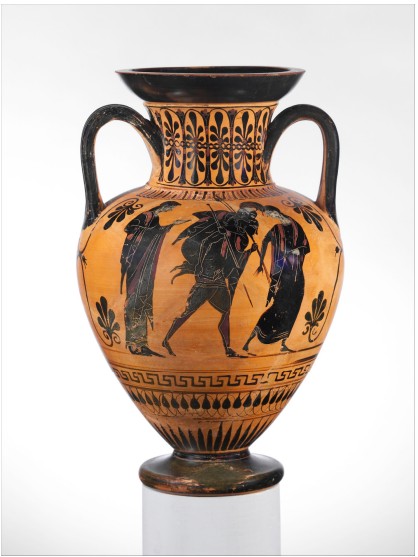

**Figure 1.** Aeneas carrying his father, Anchises. Attic black-figure amphora, last quarter of the 6th century BCE. The Metropolitan Museum of Art, New York, USA. Accession Number: 41.162.171. (Open Access–Public Domain (The Met) https://www.metmuseum.org/art/collection/search/2543 42 accessed date: 2 January 2023).

The scene has been imagined in multiple forms and mediums over the centuries, appearing on ceramic vases, marble friezes[23] and wall paintings[24] and shaped into objects such as this terracotta figurine from the 1st century CE, found in Pompeii, Italy (Figure 2).

The popularity of the image, however, was not motivated by an interest in the plight of those escaping conflict and needing to seek refuge. Rather, its reproduction was driven by Greco-Roman cultural trends of depicting scenes from the Trojan legend. This Aeneas scene became ubiquitous, especially from the Roman Late Republican period onwards, with the rising fame of Vergil's epic *Aeneid* and the story's appropriation to serve as a founding myth of Rome, which was rapidly becoming the head of a growing empire.

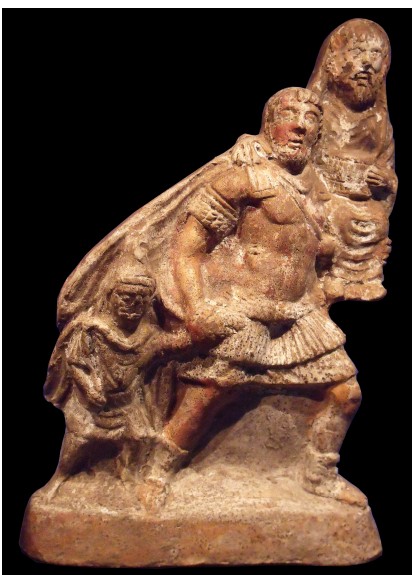

**Figure 2.** Aeneas carrying his father, Anchises, and his son, Ascanius. Terracotta figurine from Pompeii, 1st century CE. National Archaeological Museum, Naples, Italy. Accession Number: Inv. 110338. Photo Source: Alphanidon. (Creative Commons Licence: CC BY-SA 3.0. https://upload. wikimedia.org/wikipedia/commons/2/2a/Terracotta_Aeneas_MAN_Naples_110338.jpg accessed date: 2 January 2023).

A different kind of scene construction, which is familiar from narratives of supplication, is centred on sanctuaries, altars and statues of the gods: the inviolable sites of divine protection from where appeals were made. These sites were intended to shield suppliants from violence and to prevent their eviction or return to a state of danger (perhaps in the spirit of today's nonrefoulement clause in Article 33 of the UN 1951 Geneva Convention on refugees).[25] They were also there to stop perpetrators from gaining access to the suppliants. Historically, we know that sanctuaries performed these short-term refuge roles and that at times their protective powers were wanting – in that suppliants were driven out and their appeals ignored.[26] What is absent from the historical record, however, is any detailed insight into how negotiations progressed between suppliants, who lodged at such sites, and their potential hosts. The historic plea of the Plataeans as presented by Isocrates,[27] which will form the focus of the final section of this paper, is not made from the grounds of such a sacred setting. It is situated within the institutions of the polis (city-state) and framed within its semantic register of diplomacy. Unlike the historical record, the ancient literary and visual corpus re-create such sacred spaces by drawing on mythical subjects to portray the tensions of exchange and the agonising vacillation between force and fragility, as hope grows or recedes. Such a space, thronged by sculpted gods and symbols of the divine, is brought to life by the Danaids' chorus of Aeschylus's *Suppliant Women*:

A. Chorus:

> O ancestral gods, hear us with favour, and see where justice lies:
>
> by not giving our youth to be possessed in marriage
>
> against what is proper,
>
> by showing you truly hate outrageous behaviour,
>
> you will act justly < >
>
> Even for distressed fugitives from war
>
> an altar is a defence against harm that gods respect.
>
> (Aeschylus, *Suppliant Women*, lines 79–85)
>
> . . .

B. Pelasgus (king of Argos):

> Why do you say you are supplicating me in the name of these Assembled Gods,
> holding these fresh-plucked, white-wreathed boughs?

Chorus:

> So that I may not become a slave to the sons of Aegyptus.
>
> (Aeschylus, *Suppliant Women*, lines 333–35)
>
>  . . .

C. Chorus (addressing Pelasgus):

> If you don't make a promise to our band that we can rely on—
>
>  . . .
>
> With all speed . . . [we will] hang ourselves from these gods.
>
>  . . .
>
> You understand! I have opened your eyes to see more clearly.
>
> (Aeschylus, *Suppliant Women*, lines 461, 465, 467)

By continually making reference to the protective altars and statues surrounding them, the Suppliant Women's chorus animates the divine beings and compels them into guardianship of the refuge seekers. While no ancient images depicting episodes from the *Suppliant Women* have been recognised to date (Taplin 2007, p. 146), illustrations of other such moments do survive. The most analogous is a scene on a red-figure Lucanian pelike of circa 400 BCE, from South Italy (Figure 3). The composition illustrates an episode from the myth of Herakles' children in flight from the Argive king Eurystheus, which is the subject of Euripides' tragedy *Herakleidai*.[28]

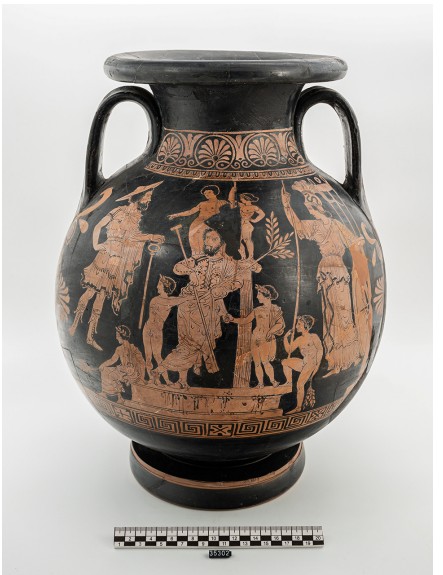

**Figure 3.** Children of Herakles, and Iolaus. Lucanian red-figure pelike, ca. 400 BCE, from near Heraclea, South Italy. Policoro, Museo Nazionale della Siritide, Italy (inv. N. 35302). (By permission from Ministero della Cultura, Direzione Regionale Musei Basilicata).

The scene depicts children within a sacred precinct scattered around an older man leaning on a column who holds suppliant branches, which confirms them as a group of suppliants. This central figure is likely Iolaus, the guardian and uncle of Herakles' children who led them to the Athenian sanctuary, from the safety of which they plead for refuge. The protective quality of the space on the image is signalled by its separation from the rest of the scene, often by the presence of a raised platform or other structures surrounding it.

At its centre it often has either an altar or, as in this case, a column or pedestal holding up a statue of a deity or other sacred object. The sculpted deity that strides atop the column here is either Apollo or Herakles. Athena is also included, protectively standing on the right to counterbalance the threatening agent of Eurystheus on the left—dressed as a herald.

The general motif is one characteristic of supplication scenarios,[29] not just those resulting from forced displacement, and can denote a last resort of pleading for divine protection when none is forthcoming from the human sphere. It becomes a particularly powerful setting for the multiple representations of Cassandra, the virgin priestess of Apollo and daughter of Priam, the defeated king of Troy. Images of this episode from the Trojan War legend survive on diverse mediums across centuries, as depicted in the following two examples: on a tondo from an Attic cup of the 440s BCE (Figure 4) and another half a millennium later as part of a 1st-century CE wall painting in Pompeii, from the House Menander (Figure 5).

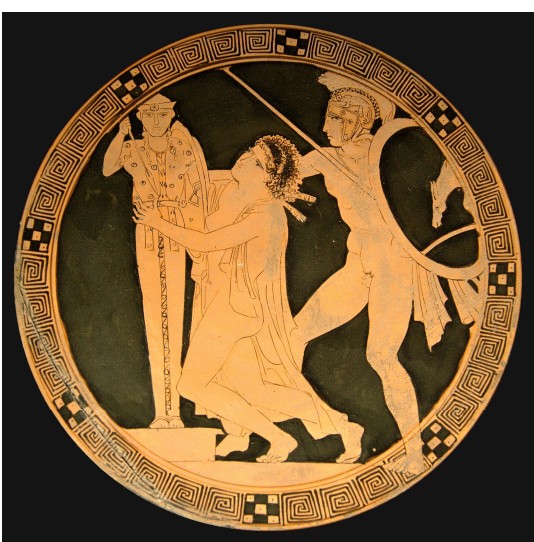

**Figure 4.** Cassandra abducted by Ajax the lesser. Tondo of an Attic red-figure cup, ca. 440–430 BCE. Louvre, Paris, France. Accession Number: Louvre G 458. (Public Domain. Image File Link: https://commons.wikimedia.org/wiki/File:Aias_Kassandra_Louvre_G458.jpg accessed date: 2 January 2023).

Portrayals of Cassandra fixate on her struggle to cling on to the seemingly unmoving statue of Athena, as the priestess is dragged away by Ajax the lesser in an act of abduction, brutality and sacrilege. The depiction of Cassandra may not itself be a direct representation of exile or displacement, but it does convey the urgent struggle for refuge and protection, which is denied by gods and men. This episode from the Trojan War epic foreshadows imminent displacement, hardship and, particularly through Cassandra's role in the story, the palpable violence already experienced in war and now to be further endured in the subjugation to come.

As with the images of Aeneas, these too depict recognisable stories from popular myths and legends, created to appeal to the latest cultural trends and specific market settings. The Lucanian pelike, with the scene of Herakles' children, was found near the ancient South Italian town of Herakleia/Heraclea (Policoro), as was another vase with a similar scene, a column-krater created around the same time.[30] The choice of image in this context may be explained by the association of the site with Herakles and hence also a regional interest in Euripides' dramatisation of it in his tragedy *Herakleidai*, which was a likely source for the images on the vases. Equally, it may be part of a more widespread growing interest in the Greek dramatic corpus, representations of which remain on a significant number of ancient vases across sites in South Italy from the 5th century BCE onwards. The site of Ruvo, for example, had the remains of some 40 such vessels (Robinson 2004, 2014, p. 326).

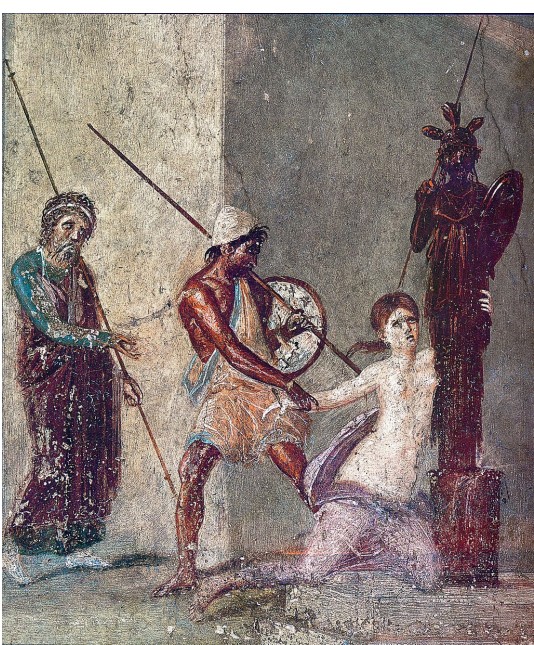

**Figure 5.** Cassandra abducted by Ajax the lesser with her father, Priam. Section of a wall painting, 1st century CE, from the House of Menander (I, 10, 4), Pompeii, Italy. Image Source: Ranieri Panetta (2005, p. 349). (Public Domain. Image File Link: https://commons.wikimedia.org/wiki/File: Pompeii_-_Casa_del_Menandro_-_Menelaos.jpg accessed date: 2 January 2023).

Although we do not know the exact provenance of the cup with the tondo of Cassandra and lesser Ajax in Figure 4, other than it was likely made in Athens and found in central Italy,[31] a very well-preserved context remains for the depiction of the scene on the Pompeian wall painting in Figure 5, which adorned the atrium of the House of Menander (I, 10, 4). The house is richly decorated with images that were likely copies of earlier now-lost paintings and mosaics familiar in Italy, which depict literary and other motifs throughout the ages (Varriale 2012, pp. 170–71). Among these were a number with episodes from the Trojan War cycle. The choice to include images of emotive and painful subjects, such as Cassandra's violent abduction and ensuing rape, may have provided potential for reflection on the suffering and brutality endured by the victims of conflict. However, from its context in this Pompeian house, it is evident that they were chosen primarily for the prominence of the image (or its artist) and perhaps the capacity of the subject matter to be particularly arresting, thus ultimately drawing the visitor's attention to the work as a cultural artefact. The ancient elite house thus became a gallery displaying the owners' cultural appreciation, cosmopolitan knowledge, connectivity and wealth, demonstrated by their capacity to have the means to acquire such objects or commission artisans to create them.

Looking at these images through the lens of the theme of this volume, *Refugees and Representation*, abstracted from the context of their creation, concentrates the gaze on the convergence of human suffering and the importance of the individual story through which it is conveyed. In contemplating them, we may perceive that their subjects challenge the observer to look beyond the pathos of the victims towards their persistent strength and the human will needed to endure and overcome conditions beyond their control. It may also allow contemplation on the way that such circumstances are not separate from but instead part of the society that is implicated in their making.[32] While such images may have evoked similar reflections from the ancient viewers, the choice to reproduce these dramatic scenes, whether on ceramics or on wall paintings, was based on their capacity to display affinity with a particular sociocultural knowledge system and, potentially, political affiliations, whether that of Pan-Hellenism or Imperial Rome.

## 4. Modes of Petitioning in Negotiations for Refuge and Protection

Surviving ancient representations of historical persons or mythical figures who are exiled or in need of protection and refuge are not generally presented as the 'other'. They are portrayed as part of the wider sociocultural (if not always geopolitical) thought world and their circumstances the result of its creation. Even those protagonists who arrive from elsewhere to seek refuge, by speaking on their own behalf (as does the chorus of Aeschylus' *Suppliant Women*), preclude the possibility of being othered despite being newcomers. In telling their own story, they convey the multiplicity of lifeways and circumstances that brought them into such a predicament. Their bodily and speech acts are forcefully conveyed even from within a condition of precarity and victimisation. Such a stance challenges the image of those seeking refuge as merely destitute, or weak and helpless.

Contemporary scholars, such as historian Gerawork Gizaw and anthropologist Liisa Malkki, who focus on current requests for asylum and contexts of refuge, recognise how pervasive and potentially damaging such tropes of the idealised deserving subject of pity and charity are (Gizaw 2022; Gizaw and Reed 2022; Malkki 1995, 1996, as well as: Behrman 2016; Cabot 2019; Coutin and Vogel 2016; Kushner 2006; Marfleet 2007; Reed and Schenck 2023). Heath Cabot, in a critical reflection on anthropology's approach to refugee issues, also recognises in her own work a 'tendency to aestheticize certain "tragic" aspects of asylum processes, which demonstrates how tropes of victimhood also imbued my own writing even as I sought to critically contest them' (Cabot 2019, p. 266). The paradox of the necessity for refuge seekers to be represented as unthreatening and deserving of sympathy, while not simply being characterised as victims, is captured within the following ancient episodes of exile and negotiations for protection. In the first extract, from the *Suppliant Women*, Aeschylus foregrounds this paradox of the suppliant's position in the dialogue between the 50 Danaids and their father, Danaeus, who counsels them as they prepare to make their case to the Argives:

> answer the natives in words that display respect, sorrow and need,
>
> as it is proper for outsiders to do,
>
> explaining clearly this flight of yours which is not due to bloodshed.
>
> Let your speech, in the first place, not be accompanied by arrogance,
>
> and let it emerge from your disciplined faces and your calm eyes
>
> that you are free of wantonness . . .
>
> Remember to be yielding—you are a needy foreign refugee:
>
> bold speech does not suit those in a weak position.
>
> (Aeschylus, *Suppliant Women*, lines 192–199)

Yet it is precisely bold speech that we hear from suppliants in Greek tragedies, such as the argument presented by the guardian Iolaus, in Euripides' *Herakleidai*. His response is addressed to the Athenian citizens and most directly to Demophon, their king. The response acts as a counter to the threatening words of the herald of the Argive king, Eurystheus, whose mission is to prevent the Athenians from giving refuge to Herakles' children:[33]

> Herald of Argive Eurystheus addressing the Athenians:
>
> I am an Argive myself, and those I am seeking to remove are Argives who have run away from my own country, persons sentenced to die in accordance with that country's laws. We, who are the city's inhabitants, have the right to pass binding sentences against our own number.
>
> (Euripides, *Herakleidai*, lines 139–46)
>
> . . .
>
> Iolaus addressing the Athenian king, Demophon:
>
> My lord, since this is the law in your land, I have the right to hear and be heard in turn, and no one shall thrust me away before I am done, as they have elsewhere.

> We have nothing to do with this man. Since we no longer have a share in Argos, and this has been ratified by vote, but are in exile from our native land, how can this man rightfully take us off as Mycenaeans, when they have banished us from the country? We are now foreigners. Or do you think it right that whoever is banished from Argos should be banished from the whole Greek world?
>
> (Euripides, *Herakleidai* lines 181–90)

The tension that pervades these speeches in the tragic corpus is equally evident in the second set of extracts from the historic case of the failed plea by the Plataeans to the Athenians in the 370s BCE, following the Theban capture of their city Platea. Their request for refuge and protection, which was addressed to the Athenian Assembly, is recounted by Isocrates in his 14th speech, *Plataicus*:[34]

> For when the Argives came to your ancestors and implored them to take up for burial the bodies of the dead at the foot of the Cadmea, your forefathers yielded to their persuasion ... and thus not only gained renown for themselves in those times, but also bequeathed to your city a glory never to be forgotten for all time to come, and this glory it would be unworthy of you to betray. For it is disgraceful that you should pride yourselves on the glorious deeds of your ancestors and then be found acting concerning your suppliants in a manner the very opposite of theirs.
>
> (Isocrates, *Plataicus*, 14.53)

> to have no refuge, to be without a fatherland, daily to suffer hardships and to watch without having the power to succour the suffering of one's own, why need I say how far this has exceeded all other calamities?
>
> (Isocrates, *Plataicus*, 14.55)

> Alone of the Greeks you Athenians owe us this contribution of succour, to rescue us now that we have been driven from our homes. It is a just request, for our ancestors, we are told, when in the Persian War your fathers had abandoned this land, alone of those who lived outside of the Peloponnesus shared in their perils and thus helped them to save their city. It is but just, therefore, that we should receive in return the same benefaction which we first conferred upon you.
>
> (Isocrates, *Plataicus*, 14.56–57)

The Plataeans do not hide their position of weakness, destitution or expulsion from their city. Instead, they confront the Athenians as equals, as previous allies, as hosts and as compatriots in exile; after all, the people of Athens, too, have endured its hardship. In praising them for their past glory and just actions, the Plataeans challenge them to live up to the prominence of their ancestors and stand by their claims of greatness. They also remind the Athenians of reciprocal duties owed for the refuge they previously received among the Plataeans.

Part of making the case for protection was also increasingly the need to present the reasons for seeking refuge as being just. As Angelos Chaniotis observes, from the early 5th century BCE, supplication and requests for asylum were perceived as claims that ought to be respected not automatically but rather only after a close examination of each case, ensuring that certain conditions would be fulfilled.[35] For the potential host, it was not the individuals who were of interest so much as their cause, which underpinned the decision as to whether the suppliant's request was sanctioned. It was important, furthermore, for positioning those who were the reason for exile and to assess their potential threat towards the seekers of refuge as well as to the hosts themselves. Certainly, these ancient episodes signal the existence of protocols that framed requests and negotiations for refuge and protection, but they do not necessarily indicate that there was recourse to any overarching policy which resulted from inter- or intracommunity directives.[36] Suppliants were dear to the gods, and in that sense, there was an expectation that those with a just cause should be given protection, with a holding to account if they were mistreated. While the sway of such

divine authority may be questionable in historical cases, that it did have some power is evident from the fact that sanctuaries and altars acted as sites of refuge from which appeals were made, at least temporarily.

Ancient cases for refuge were also built on the premise of a shared commitment to sociopolitical interdependence. Thus, there are references to honouring the duties of reciprocity, whether in the past or the future, as well as pledges to provide military and other services, as mentioned by the Plataeans in their speech to the Athenian Assembly.[37] In their case, these arguments may have had some impact but in the end were not decisive. Presumably, the most compelling factors for the hosts in coming to a resolution was assessing the level of risk and any sacrifice that would be required in providing protection. This may have included the need to take up arms to repel the threat of violence from those who instigated expulsion. It may have also shifted the balance of power between internal civic factions, either challenging or supporting those who held power. Significantly, the decision could affect the wider framework of interstate relations, either adding or losing allies.

## 5. Conclusions

The mythical and historical narratives from the ancient world reveal the shared experience of exclusion, precarity and the threat of violence. Yet there is no evident homogenising characterisation of people who are seeking refuge and protection, as distinct from the inclusion of signifiers in depictions that would identify people as suppliants and in need of safety. The difficulties that people endure in such circumstances as displacement are presented as affecting a part of one's life trajectory rather than as a defining element of a particular historical or legendary figure. There is no flattening of the multiplicity of experiences into a single moment that comes to overshadow or overdetermine the person's past and their potential for a future beyond it. There is no fixating on a refugee experience which separates those who have lived it from those nonrefugee citizens who seemingly have not.[38]

The damage that can ensue from such separating out underpins Shahram Khosravi's work reflecting on Mohsin Hamid's 21st-century novel *Exit-West*:

> Hamid says that focusing on the journey is part of the othering. He avoids journeys, which in his eyes make refugees look different from non-refugees, to focus on everyday experiences that the majority of human beings share, such as love, sadness, and the will to live.

(Khosravi 2018, p. 3; Hamid 2017)

Studies increasingly demonstrate that the category of the refugee, by homogenising diverse experiences of those who are forced to leave their homes and seek protection, allows for the subjugation of asylum claimants within legal and governmental frameworks (Malkki 1995; Blommaert 2001, 2009; Kobelinsky 2015; Cabot 2019, esp. 267–68). A key element to note in any discourse between the ancient and contemporary representations of those seeking refuge, is that in the Greco-Roman context of the Mediterranean no such legal and governmental frameworks for addressing asylum claims existed. Still, as noted earlier, there were sites designated for protection under the auspices of the divine and protocols of interdependence and reciprocity that influenced intergroup relationships and negotiations for refuge. Where the ancient context provides insight for current concerns is in the need to critically address the damage of exceptionalising the refuge, even when it is well meaning.

It is a matter for another investigation to understand why there are so few remaining ancient historical representations of such moments, especially in the period of Rome's early rise to power and imperial ambitions in the Mediterranean. It would also be important to explore the way empire affects negotiations for refuge and representations of those who drive them, particularly in a world order beyond that governed by the city-state. The influence of Christianity on the changing role of the divine in providing protection and the

positioning of suppliants, especially from Late Antiquity onwards (e.g., the discourse by Marfleet 2007, 2011; Rabben 2016; Peters 2019), provides another rich area for exploration.

Historical inquiries motivated by publications such as this (*Refugees and Representation*) allow for a richer context in which to situate the current 21st-century representation of those who seek refuge. They allow for a better understanding of the power of representation in both the undermining or facilitating of requests for refuge. In this paper I have only alluded to the unliveable situation for millions today without access to rights and protection, as I have sought to expose the extent to which representations of such lives can and do affect decisions concerning their future, made by policymakers throughout the world.[39] This raises the question of who should be able to represent the lives of those seeking refuge, especially given the impact that such representations have on the actions of the electorate. It is hoped that the longue durée perspective contributes further insight into how diverse modes of self-portrayal and representation by others can offer multiple ways of understanding the predicaments of those seeking refuge and what lies behind such representations, historically and in the present day.

**Funding:** No Funding was received for this specific research.

**Institutional Review Board Statement:** Not applicable.

**Informed Consent Statement:** Not applicable.

**Data Availability Statement:** Not applicable.

**Acknowledgments:** I am grateful for the invitation to contribute to this collection, and the opportunity to explore the ancient evidence through the lens of this volume. I am thankful for the editors' and the anonymous reviewers' careful reading of earlier versions of this paper and their helpful suggestions, which have made it a much better contribution. Any remaining errors are my own. Further thanks goes to Carmelo Colelli (Director of Museo nazionale della Siritide (Policoro)) and his colleagues, who responded so kindly to my request for getting image rights for the Children of Herakles depiction on the red-figure Lucanian pelike in Figure 3. Finally, as this piece was written during my time as visiting professor at Potsdam University (funded by the DAAD), I owe an immense gratitude to the many colleagues and students who joined the discourse from within and without contexts of displacement, especially Marcia Schenck and Gera Teferra Gizaw. I am grateful for their patience and generosity, as well as their knowledge and determination towards multiple understandings of ancient and current worlds.

**Conflicts of Interest:** The authors declare no conflict of interest.

## Notes

[1] See, for example, the way that late antique terminologies and the status categorisation of noncitizens and outsiders were connected to levels of protection (Peters 2019, pp. 87–88).

[2] (Gray 2015), in particular, discusses forms of exilic discourse in his *Stasis and Stability*. See also (Dougherty 2022; Kasimis 2018; Isayev 2022).

[3] For perspectives on permanent temporariness, see (Gizaw 2022; Hilal and Petti 2018); for context and further discussion in relation to ancient wandering, see (Isayev 2021).

[4] (Arendt 1943, p. 264) (in Kohn and Feldman edition 2007). See also discussion: (Ritivoi 2019), especially p. 103.

[5] Isocrates 14, *Plataicus*. All text and translations are from: Isocrates. *Evagoras, Helen. Busiris. Plataicus. Concerning the Team of Horses. Trapeziticus. Against Callimachus. Aegineticus. Against Lochites. Against Euthynus. Letters.* Translated by La Rue Van Hook. Loeb Classical Library 373. Cambridge: Harvard University Press. 1945.

[6] The frequency of this is indicated in the case of the Acarnians and is noted below: Polybius, *Histories* 9.40.

[7] Polybius, *Histories* 1.18.6–7.

[8] Polybius, *Histories* 1.29.6–7.

[9] Polybius, *Histories* 1.61.8.

[10] A substantial collection is included in *IG* XII, 6.1.17–41 (*IG*–Inscriptiones Graecae 2000), with detailed discussions by (Engen 2010; Gray 2015; Rubinstein 2018).

[11] Xenophon, *Anabasis*; Cicero, *ad Familiares* 4.4.4; Ovid, *Tristia*.

[12] For a recent discussion of representing contemporary seekers of refuge as a 'mass', see Behrman (2016).

[13] For an overview of ancient migration and mobility, see (Baroud and Isayev 2022; Isayev 2017a, chps. 1, 10, 11).

[14] Polybius, *Histories* 9.40.

[15] See, for example, the complicated story of Polyartus, who attempted to seek sanctuary at the public hearth of Phaselis: Polybius *Histories* 30.9.

[16] Euripides, *Medea*. For a more extensive discussion, see Isayev (2021, esp. 12–16). All text and translations are from the Loeb edition: Euripides. *Cyclops. Alcestis. Medea.* Loeb Classical Library 12. Edited and translated by David Kovacs. Cambridge: Harvard University Press, 1994. vol. 1. For a detailed discussion see (Bakewell 2013; Cole 2004; Isayev 2017b; Zeitlin 1992).

[17] Aeschylus, *Suppliant Women*, Lines 69–76. All text and translations are from Aeschylus. *Persians. Seven against Thebes. Suppliants. Prometheus Bound.* Loeb Classical Library 145. Edited and translated by Alan H. Sommerstein. Cambridge: Harvard University Press. 2009.

[18] These include studies on the ancient context, such as (Garland 2014), and the more contemporary, such as (Derrida [1997] 2000).

[19] Euripides *Medea* created in 431 BCE; Euripides, *Helen* created in 412 BCE.

[20] Livy, *Ab Urbe Condita*. For an extended discussion on the subject, see Isayev 2017a. All text and translations are from Livy. *History of Rome, Volume V: Books 21–22.* Loeb Classical Library 233. Translated by B. O. Foster. Cambridge: Harvard University Press, 1929.

[21] In relation to their presentation as dregs of society, see Jewell (2019). For critical discourse on the plebeians and earlier bibliography, see (Logghe 2017; Mouritsen 2001; Purcell 1994).

[22] Vergil *Aeneid*. All text and translations are from: Virgil. *Eclogues. Georgics. Aeneid: Books 1–6.* Translated by H. Rushton Fairclough. Revised by G. P. Goold (2019). Loeb Classical Library 63. Cambridge, MA: Harvard University Press, 1916.

[23] For example, the marble Tabula Iliaca of the 1st century CE in the Capitoline Museums, Rome: Petrain (2014).

[24] For example, the 1st-century CE wall painting that used to adorn the House of M. Fabius Ululitremulus in Pompeii, IX.13.5: (Spinazzola 1953, Tav. XVII).

[25] Convention Relating to the Status of Refugees, 189 UNTS 150, 28 July 1951 (entered into force 22 April 1954).

[26] For a discussion on historical cases and the conflicting debates around giving asylum, see (Chaniotis 1996; Naiden 2006, 2014; Sinn 1993). Examples of unsympathetic treatment of suppliants, where such treatment tends to have moralistic undertones in ancient literature, catalysed many legends that speak of punishments against the perpetrators, often in the form of natural disasters: earthquakes and tidal waves are noted for example by Pausanias 7.25.1 and Thucydides 1.128.1.

[27] Isocrates 14, *Plataicus.*

[28] That the image is most likely related to the opening scenes of Euripides *Heraklaiedai*: (Taplin 2007, p. 127).

[29] For an overview of supplication scenes on ancient vases, see (Pedrina 2017).

[30] For the interest in Greek theatre outside Greece and its representation, see (Biles and Thorn 2014; Robinson 2014; Nervegna 2015; Taplin 2007, 2012).

[31] Reference and provenance: Louvre G458: https://collections.louvre.fr/en/ark:/53355/cl010270306 (accessed on 10 December 2022).

[32] They may represent what (Sassen 2014, p. 211) refers to as the 'systemic edge': 'the extreme character of conditions at the edge makes visible larger trends that are less extreme and hence more difficult to capture'.

[33] Euripides *Heraklaiedai*. See also a discussion on the play: (Burnett 1976; Tzanetou 2012, pp. 73–104). All text and translations are from Euripides. *Children of Heracles. Hippolytus. Andromache. Hecuba.* Loeb Classical Library 484. Edited and translated by David Kovacs. Cambridge: Harvard University Press, 1995.

[34] All text and translations are from: Isocrates. *Evagoras, Helen. Busiris. Plataicus. Concerning the Team of Horses. Trapeziticus. Against Callimachus. Aegineticus. Against Lochites. Against Euthynus. Letters.* Translated by La Rue Van Hook. Loeb Classical Library 373. Cambridge: Harvard University Press. 1945.

[35] (Chaniotis 1996, esp. 82–86). For contemporary perspectives on representation of refugee experiences, asylum requests and the law, see (Behrman 2016).

[36] For a discussion on accords, statutes and laws that would have impacted on the potential to be given or denied refuge and protection: (Moatti 2021).

[37] Isocrates 14, *Plataicus*, lines 45–47, 57; for a discussion on services, see (Isayev 2017b).

[38] For a discussion on ancient refuge and contexts of citizenship, see (Gray 2018), and in relation to the *metic* (resident alien), see (Kasimis 2018).

[39] Some examples of the discourse on the most recent events include (Neumann 2021, 2022). Those concerning these last decades include (Malkki 1992; Mbembe 2003, 2019; De Genova 2010; Mezzadra and Neilson 2013; Tazzioli 2015; Boano and Astolfo 2020).

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
