# Peer review of "Representation of Whom? Ancient Moments of Seeking Refuge and Protection"

_humanities, doi:10.3390/h12020023_

Round 1

Reviewer 1 Report

See the attached file of comments. Although I marked the "English language is fine" box, there are consistent comma errors throughout the paper--it doesn't generally interfere with understanding, but it should be thoroughly proofread.

Author Response

I very much appreciate the reviewers helpful comments and the time and care taken to provide suggestion that go a long way to improving the manuscript. I have addressed all the points as indicated by the Academic editors of the review. 

A small point to the publishers that all the quotes that are indented should be done along the style of the one on lines 173 - 183 - on the updated manuscript.

.. and sorry that you had to tackle my comma fiasco through the manuscript - which has now been tamed.  

Reviewer 2 Report

The paper “Representation of Whom? Ancient moments of seeking Refugee and protection” is an interesting exploration around different forms of representation in Greco roman and Mediterranean ancient documents on how refuge, protection and supplication was depicted and narrated. It well continues the series of sponsored by the Journal on refugee in the antique and I personally find interesting and well depicted. I learned a lot from the author(S), and I commend the capacity to link such disparate and diverse scholarship to infuse a critical reflection on the contemporary scenes of forced migrations, conflicts, and refuge. I do think however, the manuscript will benefit of clarification and modification to achieve a higher standard and a much effective communication to an interdisciplinarity audience. I do list comments and suggestions below, not necessary in order or priority and would leave to the editor to suggest otherwise and the author to amend the manuscript.

-       Abstract: I suggest revising completely the abstract making more accessible and not repeating the first part of the introduction. It would be important the abstract situate the paper in the research and would anticipate the main argument trying to capture the essence of the structure and the aim of the writing.

-       Representation of whom is a great question and is great to see it has used rhetorically to structure the first part of the paper. I would however suggest explaining better in the introduction maybe (but would leave the choice to the author to find a mora appropriate spot) what are the meaning of representations cutting across poetry, arts, literature and how somehow the aesthetic and the visuals are used to investigate and portrait the scenes, the landscapes and the relations of refugee seeking and how the subject of refuge is portrait. The point here is to me to make more explicit the frames and the disciplines where representation is so central and to better guide the reader

-       The negotiation of refuge does seems a central analytics but I do wonder if would be possible to make it more clear and frame it more explicitly. What constitutes such negotiation? How has it analyzed? What is the role of landscapes? Persons? And context? In shaping this?

-       I think the point on the non-genericity of such stories, protagonist event is needed and is interesting though the common situation of precarity and displacement, beyond an exceptionality, might need more sustained reflections especially from the present. What is never presented nor documented in the paper is the current global conditions of displacement and precarity both numerically as well as aesthetically connected with the complex violent border regimes. More references and connection of a wider literature might be needed to sustain this (eg: Di Cesare; Boano and Astolfo; Tazzioli; De Genova; Mezzadra and Nielson; Mbembe etc)

-       I would suggest the author to consider the restructuring of the para 2 “protagonist..” as figures taken from the ancient. The logic of figure both intellectually and theosophically respond to a much clearer methodology of listing, object, narratives, representations etc, from which some protagonists emerge and from where, the author can flag out a much flavored clear and structured “representation” of the negotiation around refuge. This figure would address a more complex palimpsest of different narratives and scenarios. A good reference for what I sugges it the work of Thomas Nail, The figure of the migrant, where the author structure, although in a deconstructive manner, elements and aspects that, once recomposed, at the end, respond to a figure (other references are found in the work of Agamben for example but not necessary useable for this topic). I strongly suggest reconfiguring the manuscript not on protagonist, but on figure to better frame the representation and their intellectual, historical and artistic contest. Such rearticulation can occur from page 4 to 16 and the current headings can actually be used to argue, comparatively and across figures some reflections on the representation of the negotiations. 

-       Conclusion for me, are misleading and reads more as justification, then conclusive. While I do think is appropriate the disclaimer on the role of Chirstianity it can be better sustained and connected with other researchers (think on Di Cesare, on Francescan solidarity, on some work of Silvia Federici for example). However what is mainly missing is a complete reassessment of the different form of negotiations emerged in the previous narrative, whose composition are needed to better flag up the argument of the research. The message, at least the one I got from the reading is that the exceptionalism is to be abandoned and that the ontological or at least complete precarity of the subject asking refuge to have to be reconsidered and such signal were already present in the ancient. The point made is to better flag the message for the present and the present scholar to interrogate the present condition considering such amazing representation investigation. 

-       A simpler structure and much simpler identification of the representation narrative selected, as compass is needed

-       English is sophisticated at time though it has repetition and obscure sentences. It will benefit of some simplification and revisions. 

-       The manuscript I have has different fonts for the in-text citation, not sure is correct, please verify

Author Response

I very much appreciate the reviewers helpful comments and the time and care taken to provide suggestion that go a long way to improving the manuscript. I have addressed all the points as indicated by the Academic editors of the volume. 

This included in particular adjustment of the Abstract, the Introduction and Conclusion, as well as references to contemporary events and scholarly engagement with it, and sign-posting/streamlining other parts of the paper. I have also gone over the grammar - and addressed where chaos reigned.

Other more expansive suggestions, primarily for future work, will be incredibly helpful as I continue to investigate the subject further. 

Thank you .